# Development of a Novel Live Attenuated Influenza A Virus Vaccine Encoding the IgA-Inducing Protein

**DOI:** 10.3390/vaccines9070703

**Published:** 2021-06-27

**Authors:** C. Joaquín Cáceres, Stivalis Cardenas-Garcia, Aarti Jain, L. Claire Gay, Silvia Carnaccini, Brittany Seibert, Lucas M. Ferreri, Ginger Geiger, Algimantas Jasinskas, Rie Nakajima, Daniela S. Rajao, Irina Isakova-Sivak, Larisa Rudenko, Amy L. Vincent, D. Huw Davies, Daniel R. Perez

**Affiliations:** 1Department of Population Health, College of Veterinary Medicine, University of Georgia, Athens, GA 30602, USA; cjoaquincaceres@uga.edu (C.J.C.); stivalis@uga.edu (S.C.-G.); lindsey.gay@uga.edu (L.C.G.); baseibert@uga.edu (B.S.); lucas.matias.ferreri@emory.edu (L.M.F.); imginger@uga.edu (G.G.); daniela.rajao@uga.edu (D.S.R.); 2Department of Physiology and Biophysics, School of Medicine, University of California Irvine, Irvine, CA 92697, USA; aartij@hs.uci.edu (A.J.); ajasinsk@hs.uci.edu (A.J.); rie3@hs.uci.edu (R.N.); ddavies@uci.edu (D.H.D.); 3Tifton Diagnostic Laboratory, College of Veterinary Medicine, University of Georgia, Tifton, GA 31793, USA; scarnaccini@uga.edu; 4Department of Virology, Institute of Experimental Medicine, 12 Acad. Pavlov Street, 197376 St Petersburg, Russia; isakova.sivak@gmail.com (I.I.-S.); vaccine@mail.ru (L.R.); 5National Animal Disease Center, USDA-ARS, 1920 Dayton Avenue, Ames, IA 50010, USA; amy.vincent@usda.gov

**Keywords:** LAIV, influenza, HA, IGIP, IgA, IgG, vaccine, natural adjuvant

## Abstract

Live attenuated influenza virus (LAIV) vaccines elicit a combination of systemic and mucosal immunity by mimicking a natural infection. To further enhance protective mucosal responses, we incorporated the gene encoding the IgA-inducing protein (IGIP) into the LAIV genomes of the cold-adapted A/Leningrad/134/17/57 (H2N2) strain (caLen) and the experimental attenuated backbone A/turkey/Ohio/313053/04 (H3N2) (OH/04*att*). Incorporation of IGIP into the caLen background led to a virus that grew poorly in prototypical substrates. In contrast, IGIP in the OH/04att background (IGIP-H1att) virus grew to titers comparable to the isogenic backbone H1att (H1N1) without IGIP. IGIP-H1att- and H1caLen-vaccinated mice were protected against lethal challenge with a homologous virus. The IGIP-H1att vaccine generated robust serum HAI responses in naïve mice against the homologous virus, equal or better than those obtained with the H1caLen vaccine. Analyses of IgG and IgA responses using a protein microarray revealed qualitative differences in humoral and mucosal responses between vaccine groups. Overall, serum and bronchoalveolar lavage samples from the IGIP-H1att group showed trends towards increased stimulation of IgG and IgA responses compared to H1caLen samples. In summary, the introduction of genes encoding immunomodulatory functions into a candidate LAIV can serve as natural adjuvants to improve overall vaccine safety and efficacy.

## 1. Introduction

Influenza A (IAV) generates 3–5 million cases of severe disease, and between 300,000 and 600,000 deaths worldwide annually [1,2]. For the U.S., influenza virus infections result in an average economic impact of $87 billion due to prophylactic, therapeutic and hospitalization costs, and missed school or workdays [3,4,5]. Vaccination is considered the first line of defense against IAV, but the ever-changing nature of these viruses make vaccines ineffective after a single season or against pandemic strains. The FDA has approved three types of influenza virus vaccines for human use: split virion or subunit inactivated influenza virus (IIV), recombinant influenza protein (RIV), and live attenuated influenza virus (LAIV) vaccines. The IIV and RIV vaccines can elicit the production of antibodies that target epitopes on the HA yet produce limited or no cellular immunity. In contrast, LAIV can elicit a combination of humoral and cellular responses by mimicking a natural infection [6,7]. Despite the inherent ability of LAIVs. to provide immunity against multiple viral targets, they are not suitable for everyone due to safety concerns in immunologically compromised individuals [7,8]. Thus, improving the safety profile of LAIVs. while maintaining efficacy would be ideal to expand the use of such vaccines in the population.

IgA responses are considered of great significance to prevent and/or control genital, intestinal, and respiratory infections, including IAV [9]. After a typical influenza infection, both IgA and IgG responses are detected at the airway mucosa with neutralizing activity against influenza. IgA, particularly secretory IgA in its multiple multimeric forms, is typically more broadly neutralizing than IgG [10]. IgA neutralizes pathogens without causing inflammation because of its inability to fix and activate the complement cascade [11]. The IgA-inducing protein (IGIP) was initially characterized in the bovine gastrointestinal associated lymphoid tissue. IGIP is highly conserved among mammals with a predicted molecular weight between ~5.1 and ~5.9 KDa [12,13]. IGIP is secreted by antigen-presenting dendritic cells in the intestinal tract and has been shown to positively regulate mucosal IgA expression. We hypothesized that incorporation of IGIP in a LAIV vaccine would better stimulate protective antibody responses. In addition, we hypothesize that incorporating IGIP into the influenza virus genome would further attenuate the virus as it has been shown with other recombinant influenza viruses carrying foreign genes. To test our hypotheses, we designed an HA gene segment with a modification to allow the expression of both IGIP and the HA protein. We chose two LAIV backbones to prepare candidate vaccines against the homologous A/California/04/2009 (H1N1) challenge virus. We utilized the caLen vaccine virus approved for human use and the experimental OH/04att vaccine virus previously developed in house [14,15]. To establish differences in safety and efficacy profiles, we utilized the DBA/2 J mouse model, which has been shown to be highly susceptible to IAV without adaptation. These studies show the potential of utilizing natural adjuvants in the context of a LAIV that improves safety of the vaccine while preserving and even improving protective responses against IAV.

## 2. Materials and Methods

### 2.1. Cells

Madin-Darby canine kidney (MDCK) and human embryonic kidney 293T cells (HEK293T) were a kind gift from Robert Webster (St Jude Children’s Research Hospital, Memphis, TN, USA). MDCK STAT1 KO cells (CCL-34-VHG) were purchased from ATCC. Cells were maintained in Dulbecco’s Modified Eagles Medium (DMEM, Sigma-Aldrich, St Louis, MO, USA) containing 10% fetal bovine serum (FBS, Sigma-Aldrich, St Louis, MO, USA), 1% antibiotic/antimycotic (AB, Sigma-Aldrich, St Louis, MO, USA) and 1% L-Glutamine (Sigma-Aldrich, St Louis, MO, USA). Cells were cultured at 37 °C under 5% CO_2_.

### 2.2. Generation of IGIP-H1 Influenza Plasmids

The IGIP protein is highly conserved among mammals and expressed by antigen-presenting dendritic cells (DCs) in the intestinal tract as a 47–54 aa protein. IGIP is thought to play a role in the regulation of IgA expression in the intestinal tract. The C-terminal 24 aa in IGIP correspond to the mature active peptide, whereas the N-terminal ~30 aa correspond to the signal peptide region (Figure 1A). DNA fragments with the sequence corresponding to the 5′ untranslated region (UTR) and signal peptide sequence of H1 HA (A/California/04/09 (Ca/04) (H1N1)), followed by a G4S linker, furin cleavage site, the Thosea assigna virus (TAV) 2A protease and the mature IGIP were generated with a cloning spacer downstream and acquired from Genscript (Piscataway, NJ, USA). The fragment was digested with AarI (Thermo Scientific, Waltham, MA, USA) and cloned into the reverse genetic plasmid pDP2002 as previously described, generating an intermediate plasmid pDP2002-IGIP [15,16]. The pDP2002-IGIP was digested with BsmBI (New England BioLabs, Ipswich, MA, USA) and the HA Ca/04, previously amplified by PCR using Phusion High-fidelity PCR master mix with GC buffer (Thermo Scientific) was cloned into the pDP2002-IGIP generating the plasmid pDP2002-IGIP-H1. The pDP2002-IGIP-H1 sequence was confirmed by Sanger sequencing (Psomagen, Rockville, MD, USA).

### 2.3. Generation of IGIP-Influenza Viruses by Reverse Genetics

The pDP2002-IGIP-H1 or the pDPHA-H1 (Ca/04) wild type plasmids were transfected with the 6 plasmids corresponding to the OH/04 attenuated-temperature sensitive ([ts + HA tag = att]) backbone previously described [16] or the cold adapted Leningrad backbone (caLen) [17]. In both cases, a plasmid expressing the Neuraminidase (NA) of Ca/04 was used. Co-cultures of 9 × 10^5^ HEK293T and 1.5 × 10^5^ MDCK cells were seeded per well in a 6-well plate. The following day, 1 µg of each plasmid was mixed with 18 µL of TransIT-LT1 transfection reagent (Mirus Bio LLC, Madison, WI, USA). The mixture was incubated for 45 min and then used to overlay the 293T/MDCK cells overnight. The next day, the transfection mixture was replaced with fresh Opti-MEM media containing 1% AB (Life Technologies, Carlsbad, CA, USA) and 24 h post-transfection, the media was supplemented with 1 µg/mL of tosylsulfonyl phenylalanyl chloromethyl ketone (TPCK) treated-trypsin (Worthington Biochemicals, Lakewood, NJ, USA). Viral stocks were generated in 10-day-old specific pathogen free (SPF) eggs. Allantoic fluids were harvested at 48 h post-infection (hpi), centrifuged, aliquoted and stored at −80 °C. Viruses were titrated by tissue culture infectious dose 50 (TCID_50_) and virus titers were established by the Reed and Muench method [18]. Viral sequences were confirmed by next generation sequencing and sanger sequencing as previously described [19].

### 2.4. In Vitro Growth Kinetics

Confluent monolayers of MDCK or MDCK STAT1 KO cells were inoculated at a multiplicity of infection (MOI) of 0.01 for each virus. Plates were incubated 15 min at 4 °C and then 45 min at 35 °C. Subsequently, the virus inoculum was removed, and the cells were washed twice with 1 mL of phosphate-buffered saline (PBS). Opti-MEM I (Life Technologies, Carlsbad, CA, USA), containing TPCK-trypsin (Worthington Biochemicals, Lakewood, NJ, USA) and antibiotic-antimycotic solution (Sigma-Aldrich, St. Louis, MO, USA), was then added to the cells (Opti-MEM-AB + TPCK). At the indicated time points, tissue culture supernatant from inoculated cells was collected for virus titer quantification. Virus RNA from tissue culture supernatant was isolated using the MagMAX-96 AI/ND viral RNA isolation kit (Thermo Fisher Scientific, Waltham, MA, USA). Virus titers were determined using a real-time reverse transcriptase PCR (RT-qPCR) assay based on the influenza A matrix gene. The RT-qPCR was performed in a QuantStudio 3 (Applied Biosystem, Foster City, CA, USA) using qScript™ XLT One-Step RT-qPCR ToughMix^®^, QuantaBio (ThermoFisher). A standard curve was generated using 10-fold serial dilutions from a virus stock of known titer to correlate quantitative PCR (qPCR) crossing-point (Cp) values with virus titers, as previously described [20]. Virus titers were expressed as log10 TCID50/mL equivalents.

### 2.5. Mouse Studies

Female and male DBA/2 J mice (5 to 6 weeks old) were purchased from Jackson laboratories (Bar Harbor, ME). Mice were randomly distributed in the number of groups depicted in Figure 2, anesthetized with isoflurane and inoculated intranasally (I.N) with 50 µL of either phosphate buffer saline (PBS) or 1 × 10^5^ TCID50/mouse of the different vaccine candidates. At 21 days post-inoculation (21 dpi), mice were boosted with the same vaccine candidate and dose. At 21 days post-boost (dpb), mice were challenged with 1 × 10^6^ TCID50/mouse (~10,000 mouse lethal dose 50) of A/California/04/2009 (H1N1) (Ca/04) mouse-adapted strain as previously described [21]. Mice were monitored along the entire course of the experiments for clinical signs at least once daily. Mice that lost ≥25% of their initial body weight (a score of 3 or higher on a 3-point scale of disease severity) were humanely euthanized. To obtain serum samples before euthanasia, mice were bled from the submandibular vein as previously described [22].

### 2.6. Hemagglutination Inhibition Assays

Serum samples were collected at 20 dpb and 14 days post-challenge (dpc) to screen for the presence of neutralizing antibodies by hemagglutination inhibition (HAI) assays as previously described [16]. Briefly, the sera were treated with a receptor-destroying enzyme (Denka Seiken, VWR, PA, USA), incubated overnight at 37 °C, and then inactivated at 56 °C for 30 min. After inactivation, the sera were diluted 1:10 with PBS and serially diluted 2-fold and mixed with 4 hemagglutination units (HAU) of virus in a 96-well plate. The virus/sera mixture was incubated 15 min at room temperature and the HI activity was determined after 45 min of incubation with 0.5% of turkey red blood cells (RBC). HI titers below ≤10 was arbitrarily assigned a value of 10.

### 2.7. Virus Neutralization Assays

The recombinant Ca/04 (H1N1) virus carrying Nano luciferase (NLuc) gene downstream PB1 was used at 100 TCID50 of per well in a 96-well plate and incubated with 1/10 serial dilutions of serum samples collected and treated as described above. The serum/virus mixture was incubated for 1 h at 37 °C and then overlayed for 15 min at 4 °C and then 45 min at 37 °C on MDCK cells seeded in a 96 well plate the day before. The serum/virus mixture was subsequently removed and 200 µL of Opti-MEM-AB + TPCK-trypsin was added, and the cells were incubated at 37 °C under 5% CO_2_ for 48 h. The virus neutralization (VN) titers were visualized by classical HA assay and NLuc assay. For the NLuc luciferase assay the Nano-Glo Luciferase Assay System (Promega, Madison, WI, USA) was utilized using a Victor X3 multilabel plate reader (PerkinElmer, Waltham, MA, USA).

### 2.8. Virus Titration

Nasal turbinates and lungs homogenates collected from mice at 5 dpc were generated using the Tissue Lyzer II (Qiagen, Hilden, Germany). Briefly, 1 mL of PBS-AB was added to each tissue together with Tungsten carbide 3 mm beads (Qiagen). Samples were homogenized for 15 min and then centrifuged at 15,000 *g* for 10 min. Supernatants were collected, aliquoted and stored at −80 °C until further analysis. Samples were titrated by TCID_50_ and virus titers were established by the Reed and Muench method [18].

### 2.9. Histopathology Examination

Lungs were collected from a representative number of mice (*n* = 4) in each group at 5 dpc for histopathological examination. Tissues were placed in 10% neutral-buffered formalin (NBF), fixed for at least 72 h, paraffin embedded and processed for routine histopathology with hematoxylin and eosin staining (HE). Lesions were subjectively scored by a pathologist blinded to the study as: none (0), mild; ≤10% (1), mild to moderate; 11–25% (2), moderate; 26–40% (3), moderate to severe; 41–60% (4) and ≥60% (5) severe, based on lesion severity and extent of inflammation. Features considered for the scoring were the following: bronchitis/bronchiolitis, alveolitis, pleuritis and vasculitis, type of inflammatory infiltrate, presence and extent of necrosis, hemorrhage, edema (interstitial and/or alveolar), fibrin/hyaline membranes, pneumocyte type 2 hypertrophy and hyperplasia and pleural mesothelial hyperplasia. For immunohistochemistry (IHC) against IAV, a polyclonal antibody anti-IAV H1N1 (Meridian Life Science; dilution 1/1500) was used. The staining was used to estimate the intensity of viral antigens. Staining intensity and distribution were subjectively scored by a pathologist blinded to the study using a scale from none (0) to large/highest positivity (5).

### 2.10. Influenza Antigen Microarray

The influenza antigen microarray was performed as previously described [23]. Serum, BALF and NW samples were diluted 1:100 in a protein array blocking buffer (GVS, Sanford, ME, USA), supplemented with E. coli lysate (GenScript, Piscataway, NJ, USA) to a final concentration of 10 mg/mL, and preincubated at room temperature (RT) for 30 min. Concurrently, arrays were rehydrated in blocking buffer (without lysate) for 30 min. Blocking buffer was removed, and arrays were probed with preincubated serum samples using sealed chambers to prevent cross-contamination of samples between the pads. Arrays were incubated overnight at 4 °C with gentle agitation. They were then washed at RT three times with Tris-buffered saline (TBS) containing 0.05% Tween 20 (T-TBS), biotin-conjugated goat anti-mouse IgA and Biotin-conjugated anti-mouse IgG (Jackson Immuno Research Laboratories, Inc., West Grove, PA, USA) were diluted 1:400 in blocking buffer and applied to separate arrays for 1 h, RT with gentle agitation. Arrays were washed three times with T-TBS, followed by incubation with streptavidin-conjugated Qdot655 (Thermo Fisher Scientific, Waltham, MA, USA) diluted 1:200 in blocking buffer for 1 h, RT. Arrays were washed three times with T-TBS and once with water. Arrays were air dried by centrifugation at 500 *g* for 5 min. Images were acquired using the ArrayCAM imaging system from Grace Bio-Labs (Bend, OR, USA). Spot and background intensities were measured using an annotated grid (.gal) file. Mean fluorescence across antigens grouped by isotypes were used for subsequent analysis. The different antigens were acquired from Sino biological (Wayne, PA, USA).

### 2.11. Graphs/Statistical Analyses

All data analyses and graphs were performed using GraphPad Prism software version 9 (GraphPad Software Inc., San Diego, CA, USA). A one-way ANOVA was performed. A P value below 0.05 was considered significant.

## 3. Results

### 3.1. IGIP-H1att and IGIP-H1caLen Viruses Show Differences in Virus Yield 

We previously developed a stable and efficacious alternative LAIV strategy for IAV carrying a ts mutation in the PB2 ORF, and ts mutations and a C-terminal epitope tag in the PB1 ORF [ts + HA tag = att]. The att strategy share some ts mutations in common with the MDV-A caA/Ann Arbor [24] and its safety, immunogenicity, and efficacy has been demonstrated in Balb/c mice and pigs [15,24,25,26]. We expanded these studies to improve the safety profile of the att candidate and test the hypothesis that IGIP would better stimulate protective antibody responses against IAV. To further these studies, we also tested the IGIP in the caLen backbone, which is currently approved for human use. The C-terminal 24 aa in IGIP correspond to the mature active peptide, whereas the N-terminal ~30 aa correspond to the signal peptide region (Figure 1A). We chose the swine IGIP mature peptide sequence, which was cloned as N-terminal tag of the HA ORF in segment 4. Specifically, IGIP was cloned downstream of the signal peptide region of the H1 HA of A/California/04/2009 (H1N1) (Ca/04) virus, followed by the G4S linker peptide, an artificial furin cleavage site, the Thosea assigna virus 2A protease sequence, the signal peptide region of Gaussia luciferase and then the mature HA ORF (Figure 1B). The reverse genetics (RG) plasmid carrying the modified IGIP-H1 HA segment was combined with the RG plasmid encoding the N1 NA of Ca/04 and 6 RG plasmids encoding the backbone of either OH/04att or caLen. As a control, isogenic viruses carrying the wild type H1 HA of Ca04 were prepared. Although IGIP-HA (H1N1) viruses were rescued in both attenuated backgrounds (Table 1), the IGIP-H1caLen virus grew poorly in both MDCK cells and in eggs in comparison to the isogenic H1caLen without IGIP, and, therefore, it was not included in subsequent analyses. In contrast, the IGIP-H1att grew to titers such as the isogenic H1att virus (Table 1), and both showed similar growth kinetics at 35 °C in MDCK cells, as well as MDCK STAT1 KO cells (Figure 1C). More importantly, serial passages of the IGIP-H1att virus showed that the modified HA segment was maintained for at least five passages in SPF eggs and MDCK cells (Table 1; Figure 1D).

### 3.2. Studies in DBA/2 J Mice Showed Improved Safety of the IGIP-H1att in Comparison to the Isogenic H1att Virus

We analyzed the safety profile of the IGIP-H1att in comparison to the isogenic H1att and H1caLen viruses in the DBA/2 J mouse model, which shows higher susceptibility to IAVs. compared to the Balb/c mouse strain [28,29]. Groups of mice (*n* = 16/group, ½ females) were inoculated with 1 × 10^5^ TCID50/mouse with either the IGIP-H1att virus, the H1att virus, the H1caLen or mock-inoculated (PBS) (Figure 2A). Although our previous studies showed that the H1att was attenuated in Balb/c mice and in pigs, this was not the case in DBA/2 J mice. The DBA/2 J mice in the H1att group showed weight loss starting on 4 dpi with rapid deterioration of clinical signs and mortality between 8–10 dpi (1 survivor out of 16, Figure 2B,C). In contrast, no clinical signs, negligible weight changes, and no mortality were observed in mice that were inoculated with IGIP-H1att or the H1caLen groups (Figure 2B,C). These results indicate significant improvement of the safety profile of the IGIP-H1att compared to the H1att virus in DBA/2 J mice, most likely due to reduce fitness of the modified segment 4.

### 3.3. Efficacy of the IGIP-H1att in DBA/2 J Mice

In order to test the efficacy of the IGIP-H1att vaccine, vaccination in a prime-boost strategy 3 weeks apart was used (Figure 2A). Mice were similarly vaccinated with the H1caLen virus. The vaccine boost produced neither clinical signs nor mortality (data not shown). Following this, the mice were challenged with 1 × 10^6^ TCID50/mouse (~10,000 MLD50) of the Ca/04 virus at 3 weeks post-boost. The IGIP-H1att- and the H1caLen-vaccinated mice were completely protected following virus challenge with neither overt clinical signs nor mortality, unlike the mock-vaccinated/challenge controls (Figure 3A,B). Consistent with these observations, virus shedding below limit of detection were observed in samples from lungs and nasal turbinates (NT) collected from a subset of mice at 5 dpc from both vaccinated/challenge groups, but not in the mock-vaccinated/challenge group (Figure 3C,D). Histopathological examination in lungs showed more severe lesions in mice from the mock-vaccinated/challenge group compared to the other groups (Table 2). These were characterized by moderate to severe random areas of necrosis, characterized by discontinuous alveolar septa replaced by brightly eosinophilic, fibrillar material (fibrin) admixed with hemorrhage, alveolar edema, karyorrhectic cellular debris, viable and degenerate neutrophils and foamy macrophages (Figure 2E). Bronchial epithelium was occasionally affected with attenuation, deciliation and single cell necrosis. In contrast, vaccinated/challenge groups presented similar patterns of lesions with mild or mild–moderate numbers of lymphocytes, plasma cells and lesser neutrophils and macrophages expanding peribronchiolar and perivascular interstitium. Bronchial epithelium was minimally affected by deciliation, and single cell drop out (Figure 3F–G). Immunohistochemical staining against IAV antigens was detected only in lungs from the mock-vaccinated/challenge group (Table 2). This was present within the nucleus and cytoplasm of bronchial epithelial cells, alveolar macrophages, pneumocytes and within necrotic cellular debris (Figure 3I). No presence of IAV antigens were observed in any of the vaccinated groups and the negative control (Figure 3J–L). These observations indicate that the IGIP-H1att is at least as effective as the H1caLen virus in protecting mice against aggressive challenge with a homologous IAV.

### 3.4. Qualitatively Different Humoral Responses Are Produced by the IGIP-H1att Virus Compared to the H1caLen Virus in DBA/2J Mice

The humoral responses produced in the IGIP-H1att- and H1caLen-vaccinated mice were analyzed utilizing serum samples obtained at 20 days post-boost (20 dpb) from a subset of four mice/group (½ females) by hemagglutination inhibition (HAI) and virus neutralization (VN) titers (Figure 4). To establish VN titers, we utilized a recombinant Ca04 (H1N1) virus carrying a chimeric PB1 with a C-terminal Nano luciferase (Nluc). Thus, VN titers are inversely proportional to the levels of Nluc activity measured at 48 hpi. In addition, IgG and IgA responses were analyzed using a protein microarray consisting of 153 HA proteins representing group 1 (H1, H2, H5, H6, H8, H9 and H11) and group 2 (H3, H4, H7, H10) subtypes. The protein array also contains 12 NA proteins corresponding to the N1, N2, and N9 subtypes, three M1 proteins, four NP proteins and one NS1 and one NS2 protein. Further, the array also contains 22 HA proteins and two NA proteins derived from influenza B viruses (IBVs) corresponding to the two major lineages (Victoria and Yamagata) as well as a single NP protein from a prototypic IBV, which serve as negative controls (not shown). Approximately ½ of the HA proteins are displayed as full length, whereas the rest correspond to the HA1 region. Details of the strain of origin, source of the protein, and presence or absence of epitope tags are provided in the Appendix A. Both HI (average of 220 vs. 170 HI titers) and VN (average of 702 vs. 660 VN titers) titers showed a trend towards improved neutralizing responses in samples obtained from the IGIP-H1att-vaccinated mice compared to the H1caLen-vaccinated mice (Figure 4A). This trend was consistent with a similar trend of the anti-H1 HA responses in the protein microarray, in which IGIP-H1att samples were on average higher than those from the H1caLen samples (Figure 4B and Appendix A). Average IgG responses were higher against the full HA than the HA1 portion for both vaccines, perhaps due to better folding of the former and/or presence of stalk antibodies. However, it must be noted that samples from the IGIP-H1att were consistently higher against pre-pandemic HA proteins compared to the H1caLen-derived samples (statistically significantly different for the HA of A/Puerto Rico/8/34 (H1N1), *p* = 0.045). IgG cross reactive responses against group 1 and group 2 HAs were significantly lower compared to anti-H1 responses (Figure 4C,D). IgG responses to the H5 HA, particularly against the full proteins in the array showed a mixed pattern, with some reacting better with samples from the H1caLen group and some with samples from the IGIP-H1att group (Figure 4C). Responses against H9 were close to background, except against the A/Hong Kong/35820/2009 HA antigen, in which the samples from the IGIP-H1att and H1caLen groups reacted similarly (Figure 4D). Responses to other group 1 HAs were low, but in those well above background, a trend was observed in favor of samples from the IGIP-H1att group (Appendix A). Cross-reactive responses against group 2 HAs were in general negligible, except for few full H3 antigens that were recognized similarly by serum samples from both vaccine groups, and responses to HA1 and full H7 antigens in which samples from the H1caLen group were more reactive (Appendix A). Both vaccine groups showed similar serum IgA response profiles against the H1 HA (Figure 5A, Appendix A). Background serum IgA levels against other group 1 HAs were observed except for the reactivity against the HA1 derived from A/duck/Hunan/795/2002 (H5N1) which was similar between the two vaccine groups and significantly higher than background (Figure 5B,C and Appendix A). Serum IgA responses against group 2 were close to background for both vaccine groups (Appendix A). Interestingly, some IgA serum samples from the IGIP-H1att group, but not from the H1caLen group, reacted with H7 HA antigens, which is in contrast to the IgG profile against group 2 HAs (Appendix A, compared to Appendix A).

To determine whether the qualitative differences described above would translate into different recall responses post-challenge, we analyzed serum samples collected at 14 dpc. Analysis of HI and VN responses indicated about 2-fold improved responses in samples from the IGIP-H1att group compared to the H1caLen group (Figure 6A). Analyses of IgG and IgA responses post-challenge revealed consistency with the post-boost profiles. On average, higher IgG responses were observed against the H1 HA in the IGIP-H1att serum samples than in the H1caLen serum samples collected at 14 dpc with statistically significant differences among most of the post-2009 antigens but not the pre-2009 antigens (Figure 6B). Furthermore, a significant difference between vaccines was observed when all the H1 antigens were combined (Appendix A). Group 1 responses showed mixed profiles, with serum samples from both vaccine groups better recognizing the full H5 than the H9 HA antigens (Figure 6C,D). Neither vaccine group was particularly efficient at recalling IgG responses against other group 1 HA antigens (H2, H6, H8 and H11, Appendix A). Interestingly, the IGIP-H1att vaccine produced higher average IgG responses against group 2 HAs, particularly against H3 and H4, whereas responses against H7 were higher than those against H3, and both vaccine groups behaved similarly (Figure 6E and Appendix A). On average, higher serum IgA responses were observed at 14 dpc in samples from the H1caLen group compared to the IGIP-H1att group with statistically significant differences in the post-2009 H1 antigens (Figure 7A–C and Appendix A). These analyses suggest that qualitative responses to influenza viruses can be influenced by the vaccine background in mice vaccinated with different LAIVs.

### 3.5. Average Higher Anti-H1 HA Mucosal IgG and IgA Responses in the IGIP-H1att Group Compared to the H1caLen Group at 14 dpc

Analyses of recall mucosal responses were established using samples from nasal washes (NW; Figure 8A,C) and BALF (Figure 8B,D) collected from both vaccine groups at 14 dpc. These analyses revealed a statistically significant increase in IgG and IgA in BALF and IgA in NW responses when samples from the IGIP-H1att group were compared to the H1caLen group (Figure 8A–D Appendix A). IgG and IgA responses were higher against the full H1 HA antigens than their HA1 regions. In addition, recall responses were highly focused against the 2009 H1 antigens, with little to no reactivity against pre-pandemic H1 HAs (Figure 8) or other group 1 and 2 HAs (not shown).

### 3.6. Humoral and Mucosal Responses against the NA and Internal Proteins Are Consistent with Anti-HA Response Patterns in the IGIP-H1att and H1caLen Groups

The serum and mucosal IgG and IgA profiles against the NA, NP, M1 and NS1 followed the patterns observed against HA responses (Figure 9 and Figure 10). Anti-NA responses were, on average, clearly above background at 20 dpb, but only in serum samples from the IGIP-H1att group and they were largely directed to the N1 subtype (Figure 9A), whereas those from the H1caLen had background responses. Anti-NA responses, specifically against N1, increased in the IGIP-H1att serum samples but not in those from the H1caLen group at 14 dpc and was statistically significant (Appendix A). Both vaccine groups stimulated serum antibody responses against the NP, mostly IgG in the IGIP-H1att samples, but IgA in the H1caLen samples (Figure 9B). Interestingly, only the IGIP-H1att vaccine resulted in humoral IgG responses to other internal proteins, specifically against M1 and NS1 but not NS2 (Figure 9 and data not shown). M1 and NS1 responses were slightly increased after challenge in the IGIP-H1att group. Analyses of pre- and post-challenge serum responses combined suggest that antibodies against the internal proteins were dominated largely by anti-NP IgG in the IGIP-H1att group and by anti-NP IgA in the H1caLen group (Appendix A). It is of note that anti-NA IgA serum responses were negligible in both vaccine groups pre- and post-challenge (Figure 9C). Likewise, anti-NA mucosal IgG and IgA responses were at background levels for both vaccine groups (Figure 10A,B). Mucosal IgG and IgA antibodies were detected against the NP but not against other internal proteins in both vaccine groups (Figure 10C,D). It is of note that the anti-NP IgG response was on average higher in NW samples obtained from the H1caLen group than that from the IGIP-H1att group, but statistically significant differences were only observed against two out of the four NP antigens evaluated. As observed with the anti-NP serological responses, they were dominated by IgG in the IGIP-H1att group (Appendix A) but clearly by IgA in the H1caLen group (Appendix A).

## 4. Discussion

Although vaccination is considered the first line of defense against influenza, the effectiveness of current IAV vaccines in recent years has been less than ideal, combined overall below 50% [30,31,32]. Although LAIVs. have the potential to provide increased multidimensional and universal cellular and humoral responses, they have also been associated with poor efficacy. In addition, one LAIV for agricultural use against swine influenza was withdrawn from the market due to safety concerns regarding reassortment with human influenza viruses. In this report, we sought to improve both the safety and the efficacy profiles of LAIVs. Specifically, we sought to reduce the fitness of the HA segment, i.e., reduce its reassortment potential, while improving mucosal immunity against influenza. Thus, the HA segment of a prototypic 2009 H1N1 pandemic strain (Ca04) was modified to carry the IGIP mature peptide flanked by additional modifications and in frame with the mature HA ORF (Figure 1A,B).

We chose the IGIP modification because of its potential as a natural vaccine adjuvant. IGIP is highly conserved among mammals with a predicted molecular weight between ~5.1 and ~5.9 KDa (Figure 1A). The IGIP mature 24 aa peptide sequence is identical in bovine, swine, and ferrets. One single amino acid difference (lysine at position 32 instead of asparagine) is seen in the human IGIP mature peptide compared to the swine IGIP. The predicted mouse IGIP differs also in one amino acid compared to the swine homolog (threonine at position 40 instead of asparagine). The role of these different polymorphisms is unknown, and we speculated they would have a minor effect. Therefore, we chose to test whether the swine IGIP mature sequence can lead to modulation of immune responses in the context of LAIV backbones. IgA class switch in B cells occurs via both T-cell-dependent and T-cell-independent pathways, and the antibody targets both pathogenic and commensal microorganisms [13]. IGIP was shown to up-regulate IgA expression [12,13]. DCs in the intestinal tract are the primary source of IGIP [13]. The significance of DCs in the process of B cell class switch is well established [13]. Stimulation of human monocyte-derived DCs with CD40 L- and vasoactive intestinal peptide (VIP) leads to significant up-regulation of IGIP mRNA synthesis (~35 fold over background). Unlike the transforming growth factor beta (TGF-β)—a well-characterized effector of B cell class switch—IGIP is not maintained in a latent form and does not require additional processing for activation [13]. IGIP requires the presence of CD40 ligand (CD40 L) but not B-cell receptor (BCR) cross-linking to specifically stimulate IgA class switch on bovine B cells [12]. In contrast, TGF-β requires both CD40 L and BCR to exert its class switch activity on bovine B cells [12]. Human naïve IgD^+^ B cells can be induced towards IgA class switch and can be stimulated to produce IgA after incubation with CD40 L, IL-2, IL-10, transmembrane activator and calcium-modulator and cyclophilin ligand interactor (TACI)-Fc and either IGIP or TGF-β [33]. To our knowledge, there has been no evidence associating overexpression of IGIP with inflammatory or autoimmune diseases; however, overexpression of either APRIL, BAFF or TGF-β is associated with autoimmune diseases and cancer [34,35].

The additional modifications between the IGIP peptide and the HA ORF (G4S linker, furin cleavage site and Tav 2A protease) were introduced to help release IGIP from the mature HA and to reach the extracellular compartment. The strategy resulted in a chimeric IGIP peptide carrying a 12 aa C-terminal tail (G4S(K/R)7). The recombinant virus IGIP-H1att grew efficiently in MDCK cells, about 1 log_10_ lower than the isogenic H1att virus without IGIP (Table 1). Moreover, the IGIP-H1att virus was stable for at least five passages in eggs (Table 1). Only two mutations were identified in the HA segment of the E5 passage IGIP-H1 att virus with respect to the E1 stock virus: The first mutation L9P (t58c^non-syn^) falls within the signal peptide of the H1 HA upstream of the IGIP gene. The L9P mutation is predicted [27] to reduce the signal peptide cleavability from >0.9 (L9) in the wild type H1 HA sequence to 0.8765 (P9) in the mutant sequence. Nevertheless, the P9 mutation would still allow for a significant proportion of the IGIP peptide to be present without the N-terminal signal peptide sequence. The second mutation, t86c^Syn^, corresponds to a silent mutation within the IGIP ORF, and therefore it would appear inconsequential for its potential activity. Unfortunately, the IGIP-HA segment severely impaired the growth of the recombinant caLen vaccine virus, perhaps due to the latter containing a larger number of attenuating mutations compared to the OH/04 att backbone [17]. Nevertheless, we were able to make side by side comparisons between the IGIP-H1att, H1att, and H1caLen viruses in terms of virus growth kinetics in vitro and safety and efficacy evaluations in DBA/2 J mice. Previous studies have shown that DBA/2 J mice are 10–1,000 times more susceptible to IAV compared to C57 BL/6 and Balb/c mouse strains [35,36]. It must be noted that DBA/2 J mice are not immunodeficient and mount protective humoral responses against type A and B influenza viruses as well as other pathogens [16,37,38]. Despite previous studies in Balb/c and pigs [15,36,39] showing attenuation of different IAVs. carrying the att (ts + HA tag) modifications, such a strategy was not sufficient to attenuate the H1att virus in DBA/2 J mice. More importantly, the IGIP-H1att virus was attenuated in DBA/2 J mice as much as the control H1caLen virus (Figure 2). This observation also suggests that the IGIP modification leads to reduced fitness of the HA segment and therefore it will be less likely to reassort, although such assessment is beyond the scope of this report.

The IGIP-H1att virus was as efficient as the H1caLen in protecting mice against aggressive challenge with a homologous prototypic 2009 H1N1 strain (Figure 3). Challenge virus shedding was below the limit of detection accompanied by the absence of clinical signs in both vaccine groups. Analyses of humoral responses by different methods (HI, VN, and protein microarray) clearly showed trends of higher IgG responses in mice vaccinated with the IGIP-H1att virus, compared to those vaccinated with the H1caLen virus (Figure 4), not only against H1 HAs but also other group 1 HAs. As expected, serum IgA responses post-boost were low and mostly focused to the H1 HA with similar levels between vaccine groups (Figure 5). In a previous study, infection of mice with a wild type H7N9 IAV led to induction of antibodies against both group 1 and group 2 HAs in the absence of discernible HAI titers [40]. In this study, post-boost serum IgG responses against a panel of group 2 H7 HAs were also detected, particularly in samples from the H1caLen group. In contrast, this same H7 HA panel showed increased serum IgA reactivity using samples from the IGIP-H1att group. At 14 dpc, the recall serum IgG antibody continued with samples from the IGIP-H1att reacting more strongly to H1 HAs than samples from the H1caLen group with statistically significant differences (Figure 6). Serum IgG responses to other group 1 HAs showed a mixed pattern of relatively weaker signals compared to the H1 profiles. It is of note that post-challenge resulted in boosting of group 2 HA responses, particularly against the H7 panel, but also against H3 and H4 antigens in samples from the IGIP-H1att group. In contrast, post-challenge serum IgA responses were on average statistically higher in samples from the H1caLen group (Figure 7). The mucosal antibody responses detected in NW and BALF at 14 dpc had overall higher average signals for both IgG and IgA in samples from the IGIP-H1att group, and statistically significant differences between vaccines for the IgG in BALF and IgA in NW and BALF (Figure 8). The patterns of IgG and IgA responses against other viral proteins (N1 and NP, particularly) were consistent with those observed against HA (Figure 8 and Figure 9). Serum IgG anti-N1 NA responses, as well as anti-M1 and anti-NS1, were detected above background only in samples from the IGIP-H1att group, but not in those from the H1caLen group. In this regard, it important to note that various approaches to more universal influenza vaccines consider more conserved targets, such as epitopes on NA, M2, M1 and NP [41,42,43]. Additionally, NP modulates cellular immune response activating CD4+ and CD8+ lymphocytes providing cross-reactivity against zoonotic IAV strains [44,45]. In the context of LAIVs, it has been also shown that different NPs modulate differently the immune response conferring protection against heterologous challenge in the absence of neutralizing antibodies [46]. FLU-v, which has shown promising results in phase II in humans, suggests that understanding the role of NP antibodies and how to modulate the NP response could pave the way for the generation of more a universal vaccine [47]. In this report, we show that anti-NP responses were easily detected in serum and mucosal samples. Serum and BALF IgG dominated the response against NP in the IGIP-H1att group and in NW in the H1caLen group. Mucosal anti-NP IgA responses were on average higher in samples from the H1caLen group. It is commonly accepted that IgA responses are better at neutralizing primary viral targets such as HA, but not other viral proteins such as NP or other internal proteins (possibly even NA). In contrast, IgG responses would be better at targeting non primary targets for ADCC, complement fixation and antibody-mediated phagocytosis due to viral proteins expressed in infected cells. Thus, it is tempting to speculate that the pattern of IgA/IgG responses in samples from the IGIP-H1att group suggest a superior protective advantage compared to those from the H1caLen group. Overall, these studies strongly suggest that qualitative different immunological responses can be induced in response to different LAIV backbones and subsequent modifications. It must be noted that we have yet to establish whether the response patterns described above are due to IGIP exerting any biological functions. Assuming that IGIP is active, the results are counterintuitive, as we would have expected further enhancement of the IgA responses. Nevertheless, it is of great significance that the IGIP modification not only improved the safety profile of the att backbone, but it did so without sacrificing immunity against the HA. Although it is accepted that IGIP is important in modulating IgA responses, such activity is considered limited to the boundaries of the intestinal tract. Little is known about IGIP function in the respiratory tract and whether it can help stimulate both IgA and IgG responses. The combined analysis of the data suggests that I.N. administration of the IGIP-H1att vaccine stimulated higher systemic IgG responses and higher IgG and IgA mucosal recall responses than the H1caLen vaccine, not only against HA but also other viral antigens. Thus, it is tempting to speculate that IGIP acts as a general adjuvant in the respiratory tract that produces enhanced IgG and IgA responses. More studies beyond the scope of the present report will be needed to better understand the role of IGIP, if any, in the modulation of immune responses in the context of LAIVs. as well as other recombinant vaccine platforms.

## 5. Conclusions

We developed a LAIV carrying the IgA-inducing protein (IGIP) which improves the attenuation of our candidate LAIV maintaining the immunogenicity and protection conferred. The introduction of genes encoding immunomodulatory functions into a candidate LAIV can serve as natural adjuvants to improve overall vaccine safety and efficacy.

## Figures and Tables

**Figure 1 vaccines-09-00703-f001:**
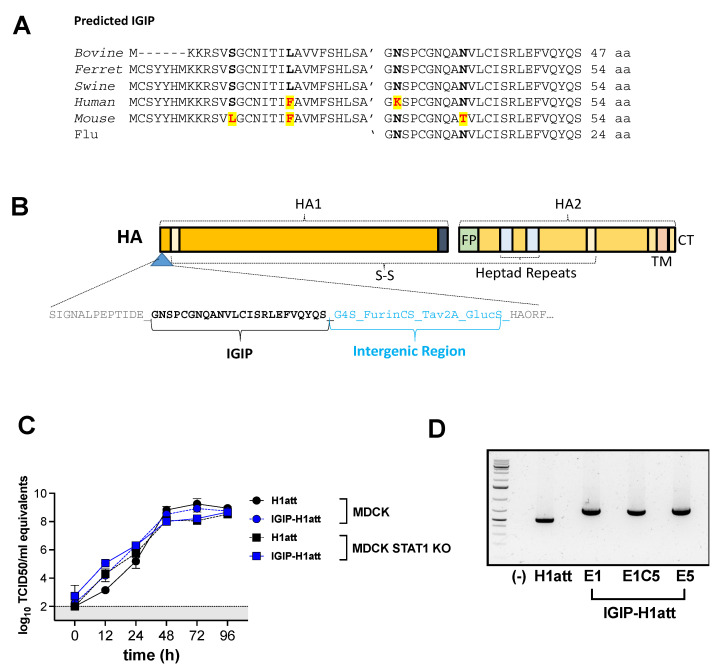
In vitro growth kinetics. (**A**) Alignment of the predicted IGIP in different mammalian species. The mature swine IGIP sequence used in this study is shown. (**B**) Schematic representation of the IGIP-H1 gene. The IGIP sequence and the components of the intergenic region are indicated. (**C**) Growth kinetics profiles of IGIP-H1att and H1att in MDCK and MDCK STAT1 KO cells. Experiments were performed two times independently, each time in triplicate. Titers were determined by RT-qPCR and expressed as log_10_ TCID50 equivalents. Gray area represents the area below the level of detection of the assay. (**D**) IGIP-H1att virus was serially passaged five times in MDCK (E1C5) cells and SPF eggs (E5), and the HA was amplified by RT-PCR, showing that the IGIP-H1 rearrangement is stable. Abbreviations: FP, fusion peptide; TM, transmembrane domain; CT, c-terminal region; G4S, poly-glycine protein linker; Furin CS, furin cleavage site; Tav2A, Thosea assigna virus 2A protein sequence; GlucS, Signal peptide of Gaussia luciferase.

**Figure 2 vaccines-09-00703-f002:**
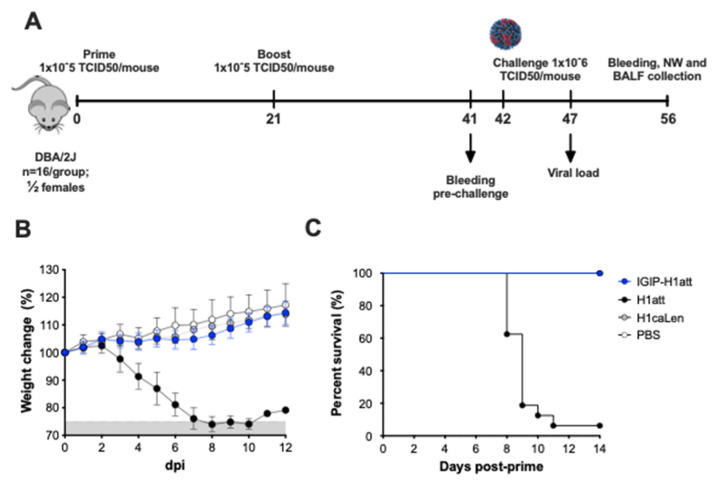
In vivo safety evaluation of IGIP-H1att, H1att, and H1caLen in DBA/2 J mice. (**A**) Schematic representation of the evaluation of the different viruses in DBA/2 J model. Mice (*n* = 16) were mock inoculated (PBS; white circles) or inoculated with 1 × 10^5^ TCID50/mouse of IGIP-H1att (blue circles), H1att (black circles) or H1caLen (grey circles). (**B**) Weight changes (grey area below the 75% mark the point where mice have reached a humane endpoint) and (**C**) Survival were monitored for 14 days after virus inoculation.

**Figure 3 vaccines-09-00703-f003:**
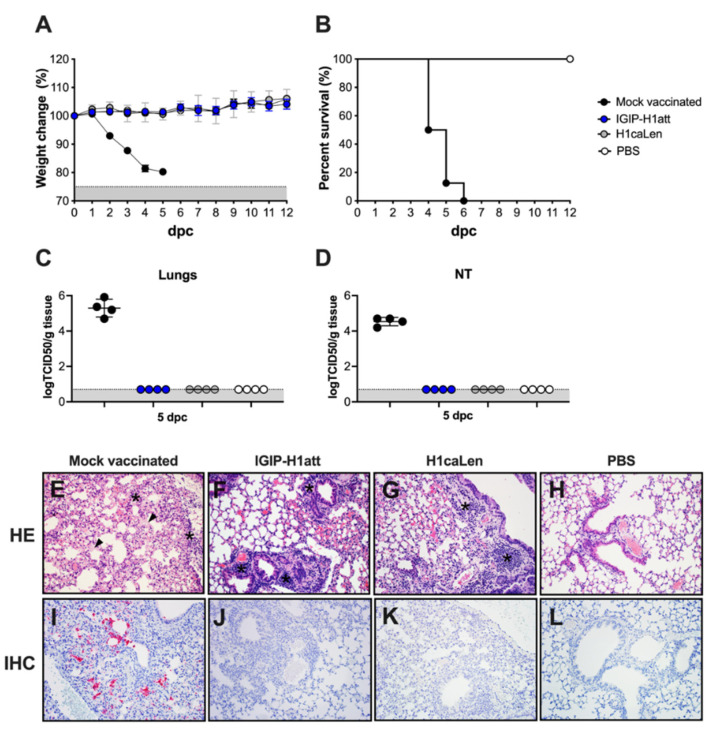
Efficacy of IGIP-H1att against H1N1 lethal challenge in DBA/2 J mice. Mice (*n* = 12/group) previously mock-vaccinated were mock challenged (white circles) or challenged (black circles) with 1 × 10^6^ TCID50/mouse of Ca/04 (H1N1). Mice previously vaccinated with IGIP-H1att (blue circles) or H1caLen (grey circles) were challenged similarly. (**A**) Weight changes (Grey area represents mice reaching humane endpoints) and (**B**) Survival were monitored for 12 days. At 5 dpc, mice (*n* = 4/group) were humanely euthanized, and the viral load was evaluated in tissue samples from (**C**) lungs and (**D**) nasal turbinates. Gray area corresponds to levels below detection for the assay (**E–H**) Histopathological examination from lungs collected at 5 dpc from each group. (**E**) Mock-vaccinated group: Multifocally the alveolar septa are necrotic, collapsed, ruptured and thickened by hyaline membranes (arrowhead). Small to moderate numbers of macrophages, neutrophils, and lesser lymphocytes and plasma cells are infiltrating perivascular and peribronchial spaces, alveolar septa, and pleura (asterisks). (**F**) IGIP-H1att- and (**G**) H1caLen-vaccinated groups: Small multifocal clusters of lymphocytes and plasma cells are infiltrating peribronchial and perivascular spaces (asterisks). (**H**) Normal lung. (**I–L**) Immunohistochemistry against IAV antigens in lungs collected at 5 dpc. (**H**) Mock-vaccinated group: Moderate amount of IAV antigens is present as evidenced by the red staining. (**J**) IGIP-H1att, (**K**) H1caLen and (**L**) PBS groups. All representative pictures were taken at 20× magnification.

**Figure 4 vaccines-09-00703-f004:**
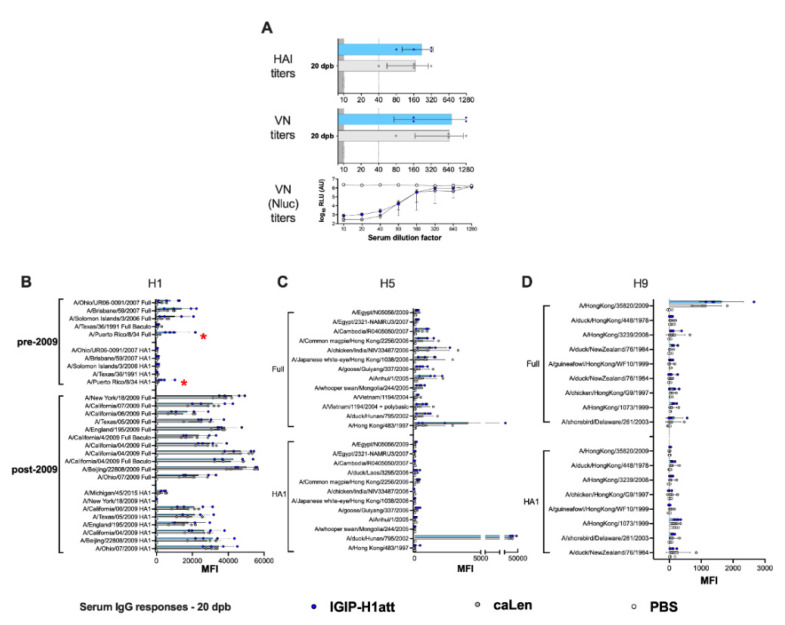
IgG serum responses 20 dpb in IGIP-H1att- and H1caLen-vaccinated mice. Mice (*n* = 4/group) were bled at 20 dpb, and the sera were used to evaluate HAI, VN and antibody reactivity against a panel of influenza antigens printed on a microarray. Samples from IGIP-H1att- and H1caLen-vaccinated mice indicated by blue dots/bars and grey dots/bars, respectively. PBS control samples are shown as white dots/bars. (**A**) HAI and VN titers. VN titers were established using a recombinant Ca/04 virus carrying PB1-Nluc and evaluated by two independent methods at 48 hpi using a classical HA assay and Nluc activity. Levels of IgG antibodies against (**B**) H1, (**C**) H5 and (**D**) H9. The reactivity of each serum sample against each antigen is shown by dots/antigen and the results are expressed as the mean of fluorescent intensity (MFI) of each value ± SD. The statistically significant differences between IGIP-H1att and caLen are depicted with red asterisks.

**Figure 5 vaccines-09-00703-f005:**
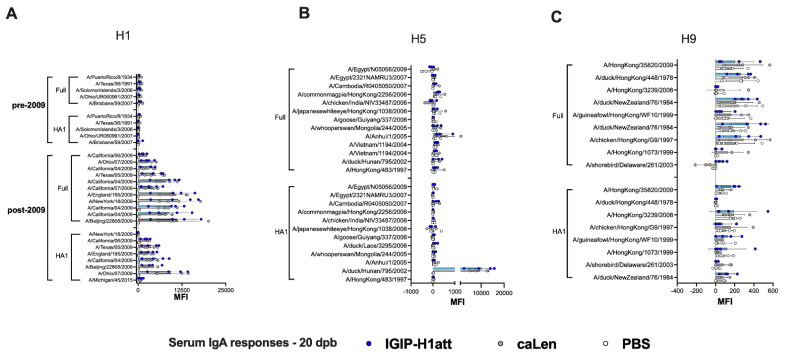
IgA serum responses 20 dpb in IGIP-H1att- and H1caLen-vaccinated mice. Same set of samples from Figure 4 probed for IgA antibodies against (**A**) H1, (**B**) H5 and (**C**) H9 using the influenza antigen array. The reactivity of serum samples is expressed as described in Figure 4 and results shown as MFI ± SD. IGIP-H1att samples in blue dots/bars. H1caLen samples in grey dots/bars. PBS control samples in white dots/bars. No significant differences were observed.

**Figure 6 vaccines-09-00703-f006:**
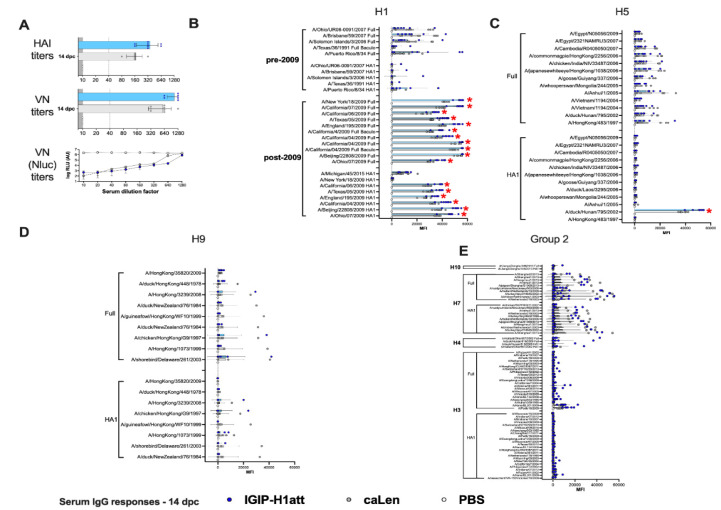
IgG serum responses 14 dpc in IGIP-H1att- and H1caLen-vaccinated mice. Mice (*n* = 8/group) were bled at 14 dpc and the sera were collected and used to evaluate antibody titers, as described in Figure 4. IGIP-H1att samples in blue dots/bars. H1caLen samples in grey dots/bars. PBS control samples in white dots/bars. (**A**) HAI, VN, and VN Nluc titers. Levels of IgG antibodies against (**B**) H1, (**C**) H5, (**D**) H9 and (**E**) group 2 HAs. The reactivity of serum samples in main graphs and insets is expressed as described in Figure 4 and results are shown as MFI ± SD. Statistically significant differences between IGIP-H1att and caLen are marked with red asterisks.

**Figure 7 vaccines-09-00703-f007:**
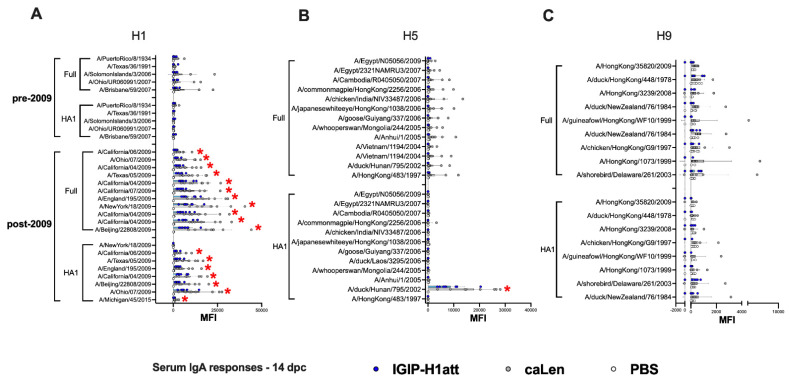
IgA serum responses 14 dpc in IGIP-H1att- and H1caLen-vaccinated mice. Same set of samples from Figure 6 probed for IgA antibodies against (**A**) H1, (**B**) H5 and (**C**) H9. The reactivity of serum samples is expressed as described in Figure 4 and results shown as MFI ± SD. The statistically significant differences between IGIP-H1att and caLen are depicted with red asterisks.

**Figure 8 vaccines-09-00703-f008:**
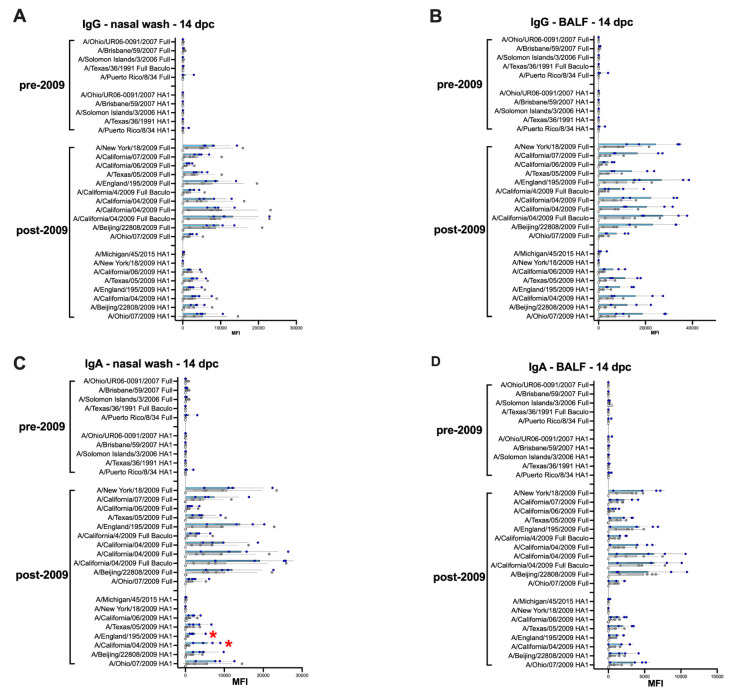
Mucosal IgG and IgA responses 14 dpc in IGIP-H1att- and H1caLen-vaccinated mice. Mice (*n* = 4/group) were humanely euthanized at 14 dpc, and nasal washes (**A**,**C**) and BALFs (**B**,**D**) were collected to evaluate the levels of IgG (**A**,**B**) and IgA (**C**,**D**) antibodies against H1 HAs on the protein microarray. The reactivity of samples in main graphs and insets is expressed as described in Figure 4 and results shown as MFI ± SD. The statistically significant differences between IGIP-H1att and caLen are depicted with red asterisks.

**Figure 9 vaccines-09-00703-f009:**
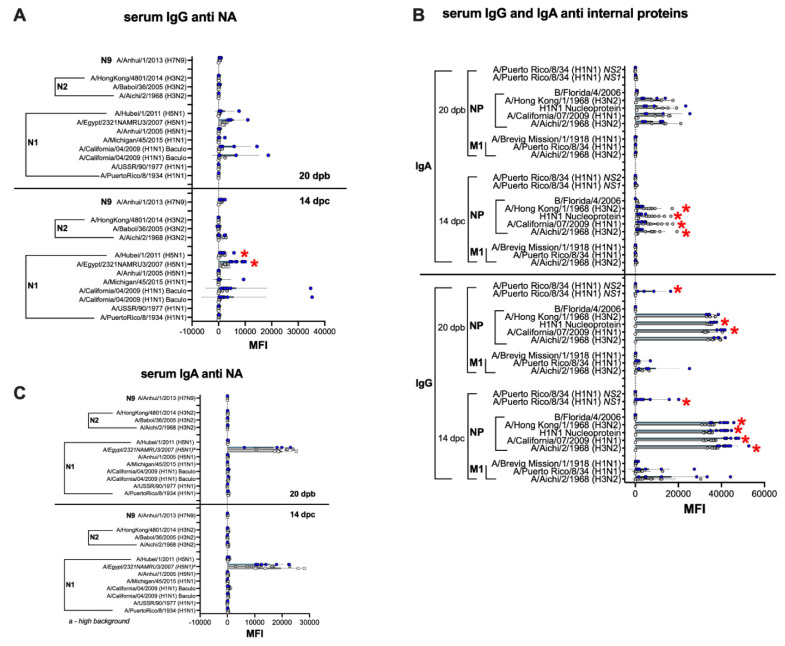
Serum antibody responses against NA and IAV internal proteins. The same set of samples described in Figure 4 (20 dpb) and Figure 6 (14 dpc) were probed for anti-NA (**A**,**C**) and anti-IAV internal proteins (**B**) antibody responses against antigens on the protein microarrays panel as indicated on the graphs. (**C**) ^a^ Data on N1 NA A/Egypt/2321NAMRU3/2007 (H5N1) antigen was not used for analyses due to high background. Statistically significant differences between IGIP-H1att and caLen marked with red asterisks.

**Figure 10 vaccines-09-00703-f010:**
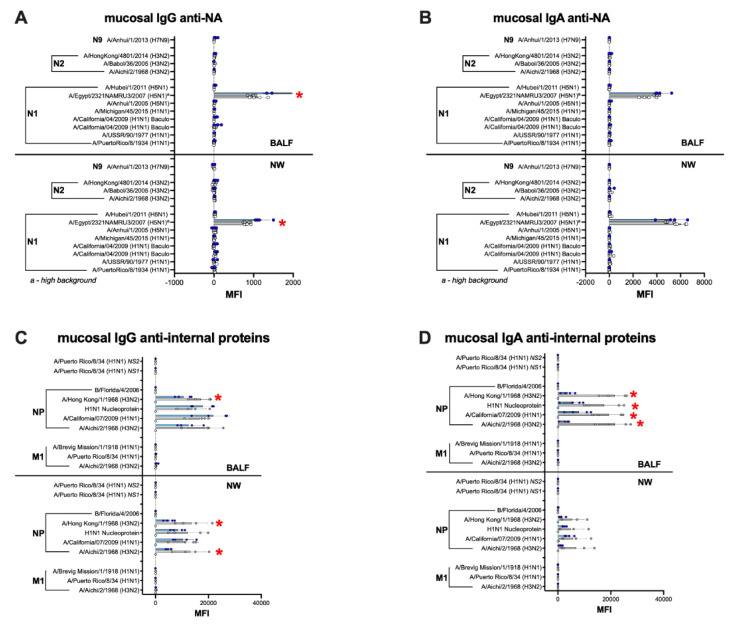
Mucosal antibody responses against NA and IAV internal proteins. The same set of samples from Figure 8 were probed for anti-NA (**A**) IgG and (**B**) IgA antibodies as well as anti-IAV internal protein (**C**) IgG and (**D**) IgA antibodies. NW, nasal washes. Insets correspond to combined BALF and NW data in which each dot corresponds to the average reactivity of each sample against anti-IAV internal proteins in the array. (**B**,**C**) ^a^ Data on N1 NA A/Egypt/2321NAMRU3/2007 (H5N1) antigen was not used for analyses due to high background. The statistically significant differences between IGIP-H1att and caLen are depicted with red asterisks.

**Table 1 vaccines-09-00703-t001:** Viral titers of the different viruses evaluated in this study.

HA Gene	Backbone	Titer TCID_50_/mL	IGIP-H1 Segment 4 Stability			
			Segment	Modification	E1 (NGS) ^a^	E5 (Sanger) ^b^
IGIP-H1	OH/04att	5 × 10^6^	HA	HA SP	No change	t58c^non-syn^(L9P) ^c^
				IGIP	No change	t86c^Syn^
				G4S	No change	No change
				Furin CS	No change	No change
				Tav2A	No change	No change
				GlucS	No change	No change
			PB2	S265 (g821)	No change	No change
			PB1	E391(g1195, g1197)	No change	No change
				G581(g1766)	No change	No change
				T661(a2005, g2007)	No change	No change
				HA tag	No change	No change
			Other segments	N/A ^d^	No change	Not performed
H1	OH/04att	2.32 × 10^7^	N/A	N/A	N/A	N/A
IGIP-H1	caLen	1 × 10^4^	N/A	N/A	N/A	N/A
H1	caLen	2 × 10^7^	N/A	N/A	N/A	N/A

Passage 1 of the virus in SPF eggs (E1) was sequenced by next generation sequencing using Illumina MiSeq. ^a^ E1 virus was sequenced by NGS ^b^ E5 virus was sequenced by Sanger sequencing of full-length HA, PB2 and PB1 RT-PCR fragments with appropriate primers (list available upon request). ^c^ L9P mutation reduces predicted signal peptide cleavability at the “…ANA-GN…” cleavage site from >0.9 to 0.8765 based on SignalP v. 5.0 predictive tool [27]. ^d^ N/A, not applicable.

**Table 2 vaccines-09-00703-t002:** Histopathological examination (HE) and immunohistochemistry against IAV scores at 5 dpc in lungs.

Group	Virus Challenge?	HE	IHC
Mock-vaccinated	Yes	4-4-4-4	3-3-3-3
IGIP-H1att	Yes	1-0-1-1	0-0-0-0
H1-caLen	Yes	1-2-0-1	0-0-0-0
PBS	No	0-0-0-0	0-0-0-0

A dash (-) separates the score of each individual mouse (*n* = 4/group, ½ females).

## Data Availability

The data discussed in this publication have been deposited in NCBI's Gene Expression Omnibus [48] and are accessible through GEO Series accession number GSE179334 (https://www.ncbi.nlm.nih.gov/geo/query/acc.cgi?acc=GSE179334).

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
