# Peer review of "Development of a Novel Live Attenuated Influenza A Virus Vaccine Encoding the IgA-Inducing Protein"

_vaccines, 2021, doi:10.3390/vaccines9070703_

Round 1

Reviewer 1 Report

In the manuscript by Caceres et al, the authors demonstrate the feasibility of incorporating the IgA-inducing protein (IGIP) in an attenuated vaccine backbone to potentially improved protective responses against IAV. The authors evaluated the efficacy of the novel vaccine using a mouse model that is highly susceptible to LAIVs including the one tested in this study. Comprehensive examination of IgG and IgA against different influenza antigens were performed. The study presented in this manuscript contributes to the efforts in improving vaccines against influenza and has merit to be published in this journal. However, there are a few inconsistencies that I am highlighting in my comments below.

Major comments:

The authors constructed a novel virus with changes in the HA ORF to incorporate IGIP. These modifications can presumably attenuate the virus given that the recombinant virus is carrying a foreign gene. However, growth kinetics conducted in MDCK or MDCK STAT KO did not provide any evidence of attenuation. Recombinant viruses H1att and IGIP-H1att had similar titers and similar level of HA expression. Attenuation of the novel vaccine could only be detected in vivo. I am wondering why the authors did not examine viremia at different stages post-prime to demonstrate that the virus is replicating in the mouse but not causing disease. This is a critical component necessary to conclude that the novel recombinant virus is attenuated in vivo. This becomes more important when suggesting that the reduced viral fitness of the novel LAVs “will be less likely to reassort” contributing to the safety profile of this product.

Mino comments:

Figure 1-3: some graphs have a gray shading that is not described in the legends as “below limit of detection”

Table 2: has a different font than the rest of the manuscript.

In figures 5-6, 8-10: some of the figures include an inset. As a reader, it is not clear to me why the inset is provided over the other sets of data. Given the amount of data presented I think is important to highlight certain sets of data but it will help the reader to understand why.

Lines 540-542: confusing sentence. Please revise.

Author Response

In name of all the authors, I would like to thank you for your comments. By responding to them and revising the ms, you have help us to improve it greatly.

Reviewer 1

Major comments:

The authors constructed a novel virus with changes in the HA ORF to incorporate IGIP. These modifications can presumably attenuate the virus given that the recombinant virus is carrying a foreign gene. However, growth kinetics conducted in MDCK or MDCK STAT KO did not provide any evidence of attenuation. Recombinant viruses H1att and IGIP-H1att had similar titers and similar level of HA expression. Attenuation of the novel vaccine could only be detected in vivo. I am wondering why the authors did not examine viremia at different stages post-prime to demonstrate that the virus is replicating in the mouse but not causing disease. This is a critical component necessary to conclude that the novel recombinant virus is attenuated in vivo. This becomes more important when suggesting that the reduced viral fitness of the novel LAVs “will be less likely to reassort” contributing to the safety profile of this product.

R: Although the observations made by the reviewer are accurate, we respectfully disagree about the interpretation of the in vitro growth kinetics. The term attenuation (att) is typically used to show the phenotype of the virus in vivo, whereas temperature sensitivity (ts) is usually demonstrated in vitro. We have already demonstrated that influenza A viruses (IAV) of avian, swine, and pandemic origin as well as influenza B viruses show a temperature sensitive phenotype at 39-41ºC when carrying ts mutations plus the HA tag at the C-terminus of PB1 (ts+HAtag=att, in the case of IAVs an additional ts mutation in PB2 is engineered). Since, modification of HA goes on top of the ts+HAtag modifications, we chose not to show the ts phenotype at 41ºC. Instead, it was important for us to demonstrate to which extent the HA modification would impair the growth of the ts+HAtag LAIV candidate at the permissive temperature of 37ºC. Regarding the analysis of viremia after prime: To our knowledge and except for highly pathogenic influenza virus strains, viremia is not a common feature of influenza viruses, much less of strains that carry attenuating mutations. Although we did not measure viral loads after prime the data clearly shows that the IGIP-H1att is significantly more attenuated in vivo than the isogenic H1att strain. We do agree with the reviewer that it will be extremely important to demonstrate whether the IGIP modification in HA makes the segment less likely to reassort. However, to demonstrate the reassortment capacity of the IGIP-HA segment, we need to design in vitro and in vivo studies involving co-infection with wild type strains. We consider such studies beyond the scope of the present report.

Mino comments:

Figure 1-3: some graphs have a gray shading that is not described in the legends as “below limit of detection”

R: We apologize for the lack of explanation. We have modified the figure legends accordingly to indicate what the grey areas represent for each graph.

Table 2: has a different font than the rest of the manuscript.

R: The font has been corrected to match the rest of the ms

In figures 5-6, 8-10: some of the figures include an inset. As a reader, it is not clear to me why the inset is provided over the other sets of data. Given the amount of data presented I think is important to highlight certain sets of data, but it will help the reader to understand why.

R: We have modified the figures by removing the insets and some of the graphs into supplementary material to improve the presentation of the results and facilitate its interpretation.

Lines 540-542: confusing sentence. Please revise.

R: The sentence has been revised and modified. The new sentence in lines 605-606 now reads: “More importantly, the IGIP-H1att virus was attenuated in DBA/2J mice as much as the control H1caLen virus (Fig 2).

Reviewer 2 Report

In this manuscript, the authors tried to improve the protective mucosal responses by incorporating the gene encoding the IgA-inducing protein (IGIP) into the LAIV genomes of the cold-adapted A/Leningrad/134/17/57 (H2N2) strain (caLen) and the experimental attenuated backbone A/turkey/Ohio/313053/04 (H3N2) (OH/04att). The IGIP in the OH/04att background (IGIP-H1att) virus grew to titers comparable to the isogenic backbone H1att (H1N1) without IGIP. Therefore, the authors used the IGIP-H1att for further analysis.

First, the authors test the efficacy of the IGIP-H1att vaccine compared with isogenic H1att and H1caLen viruses in the DBA/2J mouse model. They found a protective effect in IGIP-H1att and H1caLen compared to H1att in the aspect of weight loss and mortality. Further observations in viral titer and histopathology the authors showed that the IGIP-H1att is at least as effective as the H1caLen virus in protecting mice against aggressive challenge with a homologous IAV. Next, the authors analyzed the humoral responses by HI, VN, and protein microarray and compared them between IGIP-H1att, H1caLen and PBS groups. 

These are some of the minor issues noted.

  1. In figures 4-9, it would be helpful if the authors use a small line between groups of comparison and show the significance. It is difficult for the readers to differentiate between the sample ‘n’ or the asterisks showing significance. 
  2. Also, it would be helpful to show "non significant or NS" to indicate the groups that are not significant. 
  3. Figure 4A, can the authors clarify whether towards improved neutralizing responses in samples obtained from the IGIP-H1att-vaccinated mice compared to the H1caLen-vaccinated mice. It would be helpful if the authors provide the average mean values with standard error in the text or the data with higher "n"s. With "n" of four, it is not clear whether it is trending towards an increase.
  4. It is confusing the manuscript text and the figure legends, especially the lines 376-390. It would be helpful if the figure legends are presented in boxes to differentiate between the main text and figure legends. 
  5. Can the authors the symbol for TGF- beta?

Author Response

In name of all the authors, I would like to thank you for your comments. By responding to them and revising the ms, you have help us to improve it greatly.

Reviewer 2

  1. In figures 4-9, it would be helpful if the authors use a small line between groups of comparison and show the significance. It is difficult for the readers to differentiate between the sample ‘n’ or the asterisks showing significance. 

R: Thank you for the comment. An asterisk has been added to the groups where statistically significant differences were observed to make it easier in the identification of those specific samples

  1. Also, it would be helpful to show "non significant or NS" to indicate the groups that are not significant. 

R: Thank you, we agree with the comment. Figures have been modified to clarify where statistically significant differences are present

  1. Figure 4A, can the authors clarify whether towards improved neutralizing responses in samples obtained from the IGIP-H1att-vaccinated mice compared to the H1caLen-vaccinated mice. It would be helpful if the authors provide the average mean values with standard error in the text or the data with higher "n"s. With "n" of four, it is not clear whether it is trending towards an increase

  • The average values in each case have been added.

  1. It is confusing the manuscript text and the figure legends, especially the lines 376-390. It would be helpful if the figure legends are presented in boxes to differentiate between the main text and figure legends. 

R: The organization of the manuscripts text and legends have been modified to improve its presentation. I must note that the journal uses a template that is not easy to work with, particularly when working with the embedding of figures and tables. Our figures appear outside the margins of the main text, but if we make them smaller than they are, they become extremely hard to see and interpret.

  1. Can the authors the symbol for TGF- beta?

R: Thank you. We have corrected it accordingly.